# Butter from Different Species: Composition and Quality Parameters of Products Commercialized in the South of Spain

**DOI:** 10.3390/ani13223559

**Published:** 2023-11-18

**Authors:** Montserrat Vioque-Amor, Rafael Gómez-Díaz, Mercedes Del Río-Celestino, Carmen Avilés-Ramírez

**Affiliations:** 1Research Group AGR-120, Department of Food Science and Technology, University of Cordoba, Ctra. Madrid-Cadiz km 396, 14071 Cordoba, Spain; bt1viamm@uco.es (M.V.-A.); bt1godir@uco.es (R.G.-D.); 2Agri-Food Laboratory, CAPADR, Avda Menéndez Pidal, s/n, 14080 Cordoba, Spain; mercedes.rio.celestino@juntadeandalucia.es

**Keywords:** butter quality, color, texture, fatty acids, volatile compounds

## Abstract

**Simple Summary:**

Butter consumption has grown in recent years in Europe, and for some time now, a wide range of butters have been available on the market with a marked variability, due either to the species of origin or the manufacturing process. Applying the same analytical techniques to all, we assessed the quality of commercial butters available on the market in southern Spain. Differences were detected in nutritional value and technological parameters, many of which were linked to sensory quality. This information could be useful for the industry to learn more about the quality of each type of butter and to better promote the product.

**Abstract:**

Butter is an important product for the dairy industry due to its particular sensory attributes and nutritional value, while the variability of the composition of the fatty acids in the milk can alter the nutritional and physical properties of butter and its acceptance by consumers. Butter is highly appreciated for its distinctive flavor and aroma; however, one of its main drawbacks lies in the difficulty in spreading it at low temperatures. Several types of butter that are present in the market were used in this study. We assessed the variability in the composition of the samples regarding their texture, color properties, and volatile organic compound profiles. We analyzed samples commercially produced from sheep’s milk (SB), goat’s milk (GB), and cow’s milk (CB); samples from the latter species with (CSB) and without salt (CB); and the low-fat (CLB) version. All the physicochemical composition parameters were significantly affected by the effect of the type of butter, although only 29 out of the 45 fatty acids examined were identified in the butter samples analyzed. The textural properties of the butters were influenced by both their solid fat content and the fatty acid profile. In addition, the origin of the milk not only affected the texture parameters but also the color of the butters and the compounds associated with traits such as odor and flavor. Through the multivariate data analysis of butter fatty acids and volatile compound percentages, we observed a clear differentiation of the samples based on the species of origin.

## 1. Introduction

Butter is a product in the form of a solid, malleable emulsion, principally of the water-in-oil type, derived exclusively from milk and/or certain milk products, in which the essential constituent of value is fat, as defined by the European Union Regulation 1308/2013 [1]. This product is one of the oldest forms of preserving fat from milk, with the species of origin (cow, sheep, or goat) having an important influence on the properties of the milk [2]. The butter most commonly made is from cow’s milk, which contains essential fatty acids [3], has a distinctive and pleasant taste and aroma, and is easily digested. Cow’s milk butter remains in a solid state when refrigerated, softens to a spreadable consistency at room temperature, and melts at approximately 35 °C [4]. In contrast, butters made with sheep’s or goat’s milk have a number of different and interesting characteristics, mainly taste, aroma, appearance, and chemical composition, compared to those made with cow’s milk. In addition, sheep’s and goat’s milk are not linked to allergic reactions, and their products, due to their nutritional properties, can therefore be a valuable alternative in terms of health benefits [5].

The structure and nature of milk fatty acids and triacylglycerols are responsible for the melting point, crystallization behavior, and mechanical properties of its fat [6]. Therefore, differences in the fatty acid composition of milk fat can result in changes in the texture of butter [3], with, for instance, increased palmitic acid content making the hardness, melting temperature, crystallization temperature, and nucleation rate of the butter all increase [7].

Color intensity and a desirable flavor are the key features for consumers in assessing the quality of butter [8]. The main components responsible for the odor and flavor of many edible products, and which influence the perceived quality and customer acceptance of butter, are the volatile organic compounds (VOCs) [9]. Therefore, to broaden our understanding of perceived quality in the food industry, it is essential to incorporate technological approaches that can support the evaluation of sensory perceptions, especially because many analyses are tiring to perform or may involve health risks when repeated [10].

The flavor and texture properties of butter are influenced not only by the origin of the milk, the diet, and the stage of lactation [8], but also by additives like salt, processing treatments, and/or manufacturing practices. Visual attributes such as color, size, shape, and visual texture often contribute to the consumers’ purchasing decisions, and for butter in particular, the relevance of color and appearance for consumer acceptability has been demonstrated [10].

The characterization of butter, either by technological means or by sensory description, is an essential step in order to authenticate specific varieties and to identify abnormalities or frauds. In addition, as consumers’ demands for dairy products of high nutritional and health value have increased the popularity of small ruminants’ dairy products, it is important to increase research efforts into these valuable products. In Europe, for example, the percentage growth rate in butter production (14.22%) has increased considerably from the year 2001 to 2020 in comparison with milk production (9.28%). This growth is even more pronounced in countries like Spain, with a butter production rate of 56.14% in contrast to a milk production rate of 20.42% [11]. According to information from the Ministry of Agriculture, Livestock, and Fisheries of the Spanish Government, the per capita consumption of butter was 0.4 kg in 2021 [12]. However, very few studies have compared and characterized the overall technological quality profile of commercial butters originating from different species. Therefore, the objective of this study was to characterize and compare, using the same analytical methods, the comprehensive quality of commercial butters derived from the milk of different species in the south of Spain.

## 2. Materials and Methods

### 2.1. Experimental Material

Five commercial butters of different compositions, made from the milk of three different species, were used to perform this study. The samples were produced by the continuous method in dairy plants in areas of the province of Cordoba (Spain). The research was not a model study, but it examined the following specific, commercially available products: butter made from sheep’s milk (SB); butter made from goat’s milk (GB); and unsalted (CB), salted (CSB), and low-fat (CLB) butter made from cow’s milk. In this last low-fat version, fat was replaced mainly by water, although an emulsifier, a stabilizer, and a preservative were also added but in a percentage lower than 1%.

### 2.2. Physicochemical Analysis

Moisture content was measured by recording the weight lost from samples (initial weight of 5 ± 0.5 g) after drying in an oven at 102 ± 2 °C to a constant weight in accordance with ISO 3727-1:2003 [13]. In addition, the solid fat content (SFC), expressed as percentage by mass, was obtained by subtracting the water content and the non-fat solid content from 100, as described in ISO 3727-3:2003 [14], while the non-fat solid content was established gravimetrically after extraction of the fat from the dried butter with petroleum ether, according to ISO 3727-2:200 [15]. The salt content was measured following the Mohr method using ISO 1738:2004 [16], in which the chlorides were titrated with a silver nitrate solution in the presence of chromate anions; the titration end was indicated by the appearance of red silver chromate. The acidity values of the samples were found using the percentage of lactic acid, according to the method given in ISO 1740:2004 [17], in which samples of approximately 9 g tempered at 22 °C were titrated with 0.1 N potassium hydroxide solution using phenolphthalein as an indicator. Water activity (a_W_) was measured with a dew point hygrometer (AquaLab 4, Decagon Device Inc., Pullman, WA, USA), following ISO 18787:2017 [18]. All the measurements were taken on previously homogenized samples placed in plastic capsules and were carried out under thermodynamic equilibrium conditions.

### 2.3. Color Analyses

A Konica Minolta CR-400 colorimeter (Konica Minolta, Tokyo, Japan) was used to establish instrumental color. Values for *L** (luminosity), *a**, and *b** coordinates of the CIELAB color space were obtained. Color parameters such as hue angle (h^0^), color intensity or chroma (*C**), whiteness index (*WI*), yellowness index (*YI*), and color difference (∆*E**) were calculated according to the following formulae:(1)h0  =tan−1b*a*,
(2)$$C^{*}=\sqrt{{a^{*}}^{2}+{b^{*}}^{2}},$$
(3)$$WI=100-\sqrt{{\left(100-L^{*}\right)}^{2}+{a^{*}}^{2}+{b^{*}}^{2}},$$
(4)$$YI=142.86\cdot b^{*}/L^{*}$$
(5)$$∆E^{*}={\left[{\left({∆L}^{*}\right)}^{2}+{\left({∆a}^{*}\right)}^{2}+{\left({∆b}^{*}\right)}^{2}\right]}^{1/2}$$

Before each set of measurements, a white ceramic tile was used to calibrate the colorimeter. The color of each butter sample was then assessed at five different points, and the mean value of these measurements was taken as the color coordinate value for that particular sample.

### 2.4. Textural Analyses

The texture was analyzed at refrigeration temperature (4 °C), at an intermediate temperature (10 °C), and at room temperature (20 °C). The hardness (g) was estimated by cutting force using a wire probe (A/BC) coupled to a Stable Micro Systems TA-XT plus texture analyzer (Texture Technologies Corp., Scarsdale, NY, USA). To perform this, the probe cut a sample in the shape of a rectangular parallelepiped with dimensions of 2.5 × 10 × 2 cm to a depth of 25 mm from the surface at a speed of 1 mm/s. A penetration test was also carried out with a 6 mm cylindrical probe (P6) attached to the same equipment advancing at 1 mm/sec until it reached a depth of 5 mm. The penetration force (in g) was recorded after the probe contacted the sample and reached the minimal trigger force (3 g). The hardness was calculated as the maximum (positive) peak of resistance exerted against penetration, and the force required to retract the probe after penetration was reported as the measure of adhesiveness, which was calculated as the area of the negative region of the penetration graph (g). Each measurement determination was performed in triplicate.

### 2.5. Fatty Acid Composition Analysis

Samples (10–50 mg) were processed by adapting the one-step methylation method using chloroform described by Sukhija and Palmquist [19]. After fat extraction, the methylation and esterification of fatty acids, a gas chromatograph Agilent 6890 Network GS System (Agilent, Santa Clara, CA, USA) with a flame-ionization detector (FID), an HP 7683 automatic sample injector, and a HP-88 J&W fused silica capillary column (100 m, 0.25 mm i.d., 0.2 μm film thickness, Agilent Technologies Spain, S.L., Madrid, Spain) were used to separate and quantify fatty acid methyl esters (FAMEs). The column temperature was initially set at 100 °C and increased at a rate of 3 °C/min until it reached 158 °C; then, it increased by 1.5 °C/min up to 190 °C and was maintained for 15 min. The temperature was then increased by 2 °C/min to 200 °C and then by 10 °C/min to a final temperature of 240 °C, which was maintained for 10 min. The injection and detector temperatures were maintained at 300 °C and 320 °C, respectively. Hydrogen gas was used as the carrier gas, with a flow rate of 2.7 mL/min, and a split ratio of 17.7:1 was used to inject 1 μL of solution. Juárez et al. [20] provided detailed information on the method’s response linearity, recovery factor, precision, repeatability, and reproducibility. Nonanoic acid methyl ester at 4 mg/mL was used as an internal standard (≥97% purity; Sigma Aldrich Co., Madrid, Spain). The individual fatty acids were identified by comparing their retention times with an authenticated standard fatty acid mix Supelco 37 (Sigma Chemical Co. Ltd., Poole, UK). CLA isomers (cis9-trans11 and trans10-cis12) were identified by comparing their retention times with another authenticated standard (>98% purity; Matreya, LLC, Pleasant Gap, PA, USA). The fatty acid content was expressed as a percentage of the total methyl esters identified.

### 2.6. Volatile Compound Composition Analysis

Volatile compounds were extracted by headspace solid-phase microextraction, separated by gas chromatography, and their mass spectra were generated using the same equipment and conditions as described in Povolo et al. [21], with slight modifications. In brief, a sample of 10 g was introduced in a vial that was equilibrated by stirring for 5 min at 45 °C. A divinylbenzene/carboxen/polydimethylsiloxane (DVB-CAR-PDMS; 1 cm long × 110 μm diameter; Supelco, Bellefonte, PA, USA) fiber was conditioned according to the manufacturer’s recommendations (280 °C for 30 min in a GC injector). The fiber was then placed in the headspace above the sample for 30 min, while the vial was stirred at 45 °C. The fiber was then drawn into the needle and introduced into the split/splitless injector of the gas chromatograph. During the injection step, the splitless mode was applied for 3 min at a temperature of 270 °C. The oven was kept at 40 °C for 8 min, and then the temperature was increased at a rate of 4 °C/min until it reached 210 °C, which was maintained for 10 min. The interface temperature was 220 °C, and the helium carrier gas flow rate was 1 mL/min. Mass spectra were obtained as full scans from *m*/*z* 35 to *m*/*z* 270 (1.6 scans/s) with a source temperature of 200 °C and a 70 eV ionization potential. The area units under each peak were used to calculate the relative abundance of the volatile compounds in the chromatograms. Retention index (RI) values for each volatile compound were obtained using a series of n-alkanes under the same conditions. The compounds were tentatively identified by comparing their mass spectra with those in the National Institute of Standards and Technology (NIST; Gaithersburg, MD, USA) library or in the previously published literature.

### 2.7. Statistical Analysis

Three terrines were tested for each type of butter; thus, 15 samples in all (5 butter types × 3 replicates) were used to conduct one-way variance analyses using the type of butter as a fixed effect on physicochemical and color data, fatty acid percentages, groups and indexes, and volatile organic compound percentages. In addition, a multivariate analysis was performed on texture data, with the type of butter and the temperature of analysis as fixed effects. Post-hoc HSD Tukey Tests were carried out after each analysis, and all were performed with the STATISTICA, 12 v. software, StatSoft, Inc., Tulsa, OK, USA (2014). Finally, a multivariate analysis was performed using a Pearson correlation matrix on the mean values for grouped fatty acids and chemical families of volatile compounds, with the principal component analysis (PCA) command of the XLSTAT 2020.5.1 v.software (Addinsoft Inc., New York, NY, USA).

## 3. Results

### 3.1. Physicochemical Analysis

The physicochemical parameters of the samples are shown in Table 1. As expected for a standardized commercial product, the SFC values in all cases were similar to the fat percentages indicated on the label. All the physicochemical parameters analyzed were significantly affected by the effect of the type of butter.

### 3.2. Color Parameters

The color parameters *L** coordinate (lightness), *a** coordinate (red-green), *b** coordinate (yellow-blue), hue (*h*^0^), chroma (*C**), whiteness (*WI*), and yellowness index (*YI*) of the butter samples analyzed are shown in Table 2. The table also includes the color difference (∆*E**) for the three types of butter made from cow’s milk. All the color parameters were significantly affected by the type of butter (*p* < 0.001).

### 3.3. Texture Parameters

The texture parameters were measured at three different temperatures for each butter type. Table 3 shows the values for instrumental hardness, measured either by cutting force or by a penetration test, and adhesiveness. All texture parameters were significantly affected by the temperature of the analysis.

### 3.4. Fatty Acid Composition Analysis

The fatty acid profile (expressed as percentages), health indices (atherogenic and thrombogenic index), and spreadability index of the butter are presented in Table 4. Of the 45 fatty acids examined, only 29 fatty acids were identified in the butter samples. Overall, 27 of these fatty acids varied significantly (*p* < 0.05) among the butters. The fatty acids identified were allocated into three groups according to their chain length (SCFA, MCFA, and LCFA) and degree of unsaturation (SFA, MUFA, and PUFA), respectively. In all the groups, differences were observed among the butters (*p* < 0.001).

### 3.5. Volatile Compound Composition Analysis

Table 5 shows the most abundant volatiles identified, which were classified into ten chemical families. Forty-three volatile compounds were detected in samples originating from different milk types. GB samples presented the richest volatile profile with 38 compounds, while CLB presented the simplest, with 32 compounds. Ketones, acids, terpenes, and aliphatic hydrocarbons comprised at least 80% of the volatile compounds in all the samples.

The PCA was performed using fatty acid summations and families of VOCs of each commercial butter (Figure 1). Two significant PCs were retained, which accounted for 48.21% (PC1) and 29.27% (PC2) of the total variance. SB and CLB were separated from GB, CB, and CSB by PC1. Short- and long-chain fatty acids, saturated fatty acids, and the chemical family of ketones contributed positively to PC1, while all the chemical families of volatile compounds except ketones contributed negatively, together with medium-chain fatty acids and mono- and polyunsaturated fatty acids. Saturated, polyunsaturated, and long-chain fatty acids, together with alcohols, lactones, acids, aldehydes, and hydrocarbons (aliphatic and aromatic), contributed positively to PC2, while the remaining fatty acid groups and chemical families contributed negatively.

## 4. Discussion

The moisture content of the SB, GB, CB, and CSB samples was very similar, with a mean value of 14.99%, which was significantly lower than that found in the CLB samples, although the manufacturing process used in the latter was completely different from that of the former. However, all the samples analyzed complied with Spanish current regulations regarding butter production [1]. Regarding the SFC, significant differences were observed based on butter type (*p* < 0.001). SB had significantly lower SFC than CB, and both were lower than GB.

Water content and water droplet size are of major importance as they determine the quality and functionality of the butter [22]. The water content of butter is influenced by the technological process used, which aims to make water droplets be as small as possible and evenly distributed, in order to provide the butter with the desired consistency and smear value [23]. The water content also influences the crystallization of the fat phase and, in turn, the structure of the butter [24]. The strength of the crystals formed depends on the size of the water droplets and the amount of fat crystallized. Depending on the water content, interactions between the water droplets can occur, and the textural stability of the butter is consequently lost, together with its spreadability [25].

The size of milk fat globules varies with the species. Cow’s milk contains fat globules with a larger mean diameter (4.55 µm) than goat’s (3.50 µm) and sheep’s (3.30 µm) milk. Goat’s milk is characterized by the fact that over 65% of its fat globules have a diameter of less than 3 µm, compared to 45% in cow’s milk. This explains why goat’s milk fat is more easily attacked by digestive enzymes and is therefore faster digested [26]; these differences in structure can also condition the way in which water is trapped within the spaces between the fat globules [27], thus affecting the moisture and fat content of the final product. In addition, the differences in the size of the fat globules are also related to the rheological properties of the butters, which means that goat’s butter has a softer texture than cow’s butter [28]. This can be an advantage in parameters such as stability, but it can be a handicap for its transformation.

Apart from CLB (which presented more ingredients than milk fat), SB presented significantly higher values of non-fat solids and higher acidity (*p* < 0.001) than butter made from goat’s and cow’s milk. Non-fat solids include protein, lactose, vitamins, and minerals. Sheep’s milk has a higher protein content than goat’s and cow’s milk [27]. SB also presented the highest acidity values, in agreement with Çakmakçı, and Tahmas Kahyaoğlu [2]. This could be related to the higher protein content of sheep’s milk and, consequently, sheep dairy products. Not surprisingly, the natural acidity of milk is attributed to the presence of caseins, but it is also due to minerals, organic acids, and phosphates. The casein micelles of sheep’s milk, made up mainly of ß-caseins, are larger and more mineralized than those of cows. The calcium and phosphorous contents of sheep’s milk are also higher than in cow’s milk [29].

Water activity is a vital parameter that indicates stability during storage, since it is essential for controlling the growth of microorganisms and the speed of chemical reactions, which affect the shelf life of the product. In the current investigation, water activity values for the samples ranged from 0.909 to 0.986 in GB and CLB, respectively. CLB presented significantly higher a_W_ values (*p* < 0.001), which may be due to the free water retention, which occurs when the oil molecules combine with proteins; again, the explanation could lie in the manufacturing process of the CLB samples. No differences in water activity were found among the SB, GB, CB, and CSB samples.

Regarding color, the *L** parameter values of the samples ranged between 82.97 and 89.27 presented by GB and SB, considering only the species of milk. However, CLB presented the highest *L** value of all the samples analyzed. Differences in chemical composition might be the reason for these results. In both dairy products and cheese, the *L** parameter is closely related to water content [30].

The values of the red-green color parameter (*a**) were in the negative region of the color space for butters made of goat’s (GB) and sheep’s milk (SB), with mean values between −2.81 and −3.17, while those made with cow’s milk showed values in the positive region.

Carotene and riboflavin are the main pigments of milk [31]. Sheep’s milk, and mainly goat’s milk, lack β-carotene and present a whiter color than cow’s milk, which has a yellowish color [26]. These differences were evident for the *b** coordinate, which presented its lowest value (less yellow) in the SB samples, followed by the GB samples. Values four or even five times higher were found in all the butters made of cow’s milk, to which vitamin A had also been added as an ingredient, according to the label. In butters with a high β-carotene content, their oxidation during storage is the main reason for the color changes and, therefore, the loss of quality and the reduction in their shelf life [32]. The color of butter usually changes from yellow to light yellow [33]. In goat’s milk butters, due to the lack of carotenoids, this trend is not observed during storage.

There are many factors that affect the color parameters, such as the level of fat globule aggregation, which reduces brightness (*L**) if it is too high and increases the values of the *a** and *b** coordinates. Furthermore, the color and size of the fat globules in the initial cream, or the presence or absence of salt, affect the color parameters of the resulting butters [28]. The differences observed in the *C**, *h*^0^, *WI*, and *YI* indices, since they are calculated from the *L**, *a**, and *b** coordinates, reflect the changes that occur due to the factors mentioned above (level of aggregation and size of fat globules, salt content, or color of the raw material).

Color difference (∆*E**) is an essential parameter that is very useful for evaluating the color variations of foods in the context of technological processing. In this study, the color parameter was calculated to estimate the difference in color in butters made from unsalted (CB), salted (CSB), and low-fat (CLB) cow’s milk. To achieve this, we used the equation proposed by Bodart et al. [34]. Authors such as Zimbru et al. [35], Quintanilla et al. [36], and Adekunte et al. [37] established that the color difference between samples can be easily perceived by the human eye if the color difference value is greater than 3 (∆*E** > 3). If 1.5 < ∆*E** < 3, minor color differences can be distinguished by the human eye, whereas if ∆*E** < 1.5, they cannot.

The perception of color differences (Δ*E**) between samples is affected by both the observed color and the sensitivity of the eye. The color differences for the CSB and CLB samples compared to BC were 4.53 and 9.80, respectively. These differences should be easily detectable by the consumer.

Regarding the effect of temperature on the texture parameters of butters, as expected, hardness decreased when the temperature increased in all the samples. When butter from different species was compared, SB presented the lowest hardness values for the three temperatures. Furthermore, the addition of salt to butter made from cow’s milk increased the hardness values. The results for hardness using the penetration probe were lower at 4 °C and higher at 20 °C, respectively, than those published by Oeffner et al. [38] in butter made from cow’s milk. However, the conditions of the penetration test were not exactly the same, and the moisture values of the samples in our study were lower than those published by these authors. Regarding adhesiveness, the values detected in our analysis were slightly higher than those published by Pădureţ [39] in cow’s butter at 10 °C with a fat content in the same range as our samples. Adhesiveness increased with temperature in the GB, CB, and CSB samples, while it decreased with increasing temperature in the SB and CLB samples. In this case, the storage temperature directly affected the three-dimensional network, because hydrophobic interactions were enhanced at higher temperatures and, conversely, hydrogen bonds were strengthened with cooling. Temperature fluctuations induce different rheological behavior by affecting the consistency of the butter and altering the state and the solid fat content, which varies the distribution of solid and liquid glycerides and the functionality of the fat crystal network [23]. The number of crystal–crystal interactions formed within the products is related to product hardness, and post-crystallization phenomena increase the solid fat content and strengthen the fat crystal network. This explains why the shear and penetration assays performed at 4 °C needed much more energy to cut and penetrate the samples than those performed at 20 °C. It also accounts for the fact that both hardness values were significantly lower at 10 °C than at 4 °C.

The origin of the butter also affects texture by determining the total content of fat and type of predominant fatty acids (saturated or unsaturated) present. The samples with the highest fat contents (GB, CB, and CSB) presented the highest hardness values, while those with lower fat contents (SB and CLB) presented the lowest. Under regular conditions, milk fat-based products are used at room temperature (approximately 20 °C) and are stored in a refrigerator at 5 °C, which increases the solid fat content to a critical value. In addition, the addition of vegetable oils to a milk fat matrix decreases the hardness of these products. At the same temperature, butter has a higher fat content than margarine, and it appears with aggregated fat crystals (3 to 5 μm), with the solid and liquid fat uniformly distributed. In contrast, the fat crystals in margarine are larger (5 to 10 μm) and are primarily present at the interface of water droplets [25].

The CB and CSB samples presented the highest adhesiveness values and contained high fat values. In contrast, GB presented the lowest adhesiveness values but the highest fat percentage. This could be again a consequence of the size of the fat globules and their ability to maintain their spherical shape and water content.

The type of butter affected the fatty acid profile too. The content in short-chain fatty acids (SCFAs) ranged from 7.02% to 17.97% in the CLB and SB samples, respectively. We also observed that capric acid (C10:0) was the most abundant SCFA in the commercial butters analyzed. SCFAs have low melting points; thus, butters with a higher content tend to have a softer texture and are more spreadable [40].

Butters made from cow’s milk, except for the low-fat versions, had higher percentages of medium-chain fatty acids (MCFAs) than sheep’s and goat’s milk butters. This high percentage is attributable to their high lauric (C12:0), myristic (C14:0), and palmitic (C16:0) acid contents. Palmitic acid (C16:0) is the medium-chain saturated fatty acid that most affects the physical and rheological properties of butter, since it is present in the highest percentage in butter fat. Palmitic acid, due to its high melting point (63 °C), has a hardening effect on fat. The total saturated fat percentage present in the butter tested differed significantly between the samples (*p* < 0.001), while the degree of saturation ranged between 72.79% in CB and 78.51% in GB.

In all the types of butter analyzed, oleic acid (C18:1 cis 9) was the predominant unsaturated fatty acid, with the cow’s milk butters (both salted and unsalted) showing significantly higher values (*p* < 0.001) than those made from sheep’s and goat’s milk.

Regarding polyunsaturated fatty acids (PUFA), the most abundant was linoleic acid (C18:2n6c). Linoleic and linolenic acid are considered essential and must be provided by the diet because they cannot be synthesized by the body [41]. The butter made with goat’s milk (SB) presented the lowest values (*p* < 0.001).

The total value for n-6 (linolelaidic acid, linoleic acid, eicosatrienoic acid, and arachidonic acid), n-3 (α-linolenic acid), and n-9 (oleic acid and erucic acid) fatty acids was also calculated. Milk fat is probably the most complex of all edible fats, and milk fatty acid composition has been considered as the main factor influencing nutrition [42], with more than 400 different fatty acids detected in milk lipids so far [43]. Some of these fatty acids present biological, physiological, and nutritional properties that are highly beneficial for consumer health. Milk fat is one of the few food sources of butyric acid, and studies report that it is a potent inhibitor of cancer cell proliferation and fights against the formation of atherosclerosis [44]. The butters made with goat’s and sheep’s milk had similar contents of around 4%, which were in line with the percentages reported by other authors [45] and significantly higher than those presented by the other butters analyzed.

The long-chain PUFAs, especially oleic acid and linoleic acid, contained in dairy products, including butter, are essential in human nutrition because of their effects on promoting health and functional properties [46]. Oleic acid, in particular, is well known for its positive impact on human health as a source of energy, for its anticarcinogenic effect, and as one of the precursors of other long-chain fatty acids (LCFAs), with a beneficial role in reducing levels of bad cholesterol (LDL) in blood [47]. Here, the cow’s butters (both salted and unsalted) showed significantly higher values of these PUFAs than the other butters analyzed (*p* < 0.001).

The fatty acid profile of the butter samples was similar to that found in milk from the respective species [48]. All the butter formulations had higher amounts of palmitic acid (C16:0, 27.97–36.48), followed by oleic acid (C18:1, 14.95–19.09%), stearic acid (C18:0, 10.93–21.81%), myristic acid (C14:0, 7.02–12.29%), and lauric acid (C12:0, 2.81–4.75%). Fat intake is closely linked to cardiovascular diseases (CVD), and authors such as Kris-Etherton and Krauss [49] suggested that the replacement of saturated fatty acids with unsaturated fatty acids reduces CVD events. Thus, butter should be consumed in moderation [47]. However, it has been shown that not all SFAs are atherogenic, as is the case for stearic or butyric acid [50], and that, as has been pointed out, several PUFAs have positive properties for human health [51]. However, the atherogenic and thrombogenic indices reported by Ulbritch and Southgate [52] are dietary risk indices for cardiovascular diseases, and the consumption of dairy products with lower atherogenic and thrombogenic indices is more advisable for consumer health than those with high atherogenicity and thrombogenicity [53].

The atherogenic index (AI) indicates the ratio between fatty acids with proatherogenic properties and fatty acids with antiatherogenic properties. This index is based on the ratio of the content of fatty acids, which can increase serum cholesterol levels, such as lauric acid (C12:0), myristic acid (C14:0), and palmitic acid (C16:0) to the content of fatty acids with protective action (MUFA and PUFA, with the exception of the trans forms) [52]. Here, the AI ranged between 3.54 for CLB and 3.80 for GB. According to Yurchenko et al. [54], the consumption of foods with a low atherogenic index can lower total blood cholesterol levels.

The thrombogenic index (TI) shows the ratio between prothrombogenic (saturated) and antithrombogenic (unsaturated) products and indicates the tendency for blood clot formation [51,54]. Here, the value of this index ranged from 4.01 to 5.77 in SB and CLB, respectively. If we consider only the origin of the milk, the butters made with sheep’s milk presented significantly higher IA and TI values than those presented by goat’s and cow’s butters, which presented very similar values. However, Aguilar et al. [55] reported higher AI and TI values in goat cheeses than in sheep and cow cheeses.

The ratio of C16:0 to C18:1, which has been used in the past as an index of the spreadability of butter, was highest in the CLB samples, but when considering the origin alone, goat’s milk butter showed a better spreadability index.

Concerning the effect of butter type on VOCs, ketones represented the most abundant chemical family identified in all the butters, especially in SB samples, where 91.63% of the volatile profile consisted of ketone compounds, with 2-Heptanone and 2-nonanone being the main VOCs detected in SB. These ketones are lipid-derived compounds that usually increase with storage time [56]. Ketones such as acetone and 3-hydroxy 2-butanone (acetoin) were much more abundant not only in CB and CSB but also in GB samples. It is believed that the primary source of acetone is directly derived from the cows’ diet [57], while acetoin is derived from citrate degradation. Although diacetyl, the key compound for the typical butter taste [21], was not detected in any of the samples in this study, some of the reduction metabolites also produced during its citrate metabolism were present, such as acetoin and 2,3-butanediol.

Terpenes were the main compounds in butters made from cow’s milk, particularly in the CB and CSB samples. These compounds are well-known components of plant essential oils [58] and might be transferred directly from forage to dairy products [59]. α-Pinene was the most abundant terpene detected in the GB samples, while limonene and p-cymene were the most abundant in butters made from cow’s milk. All of these have been previously detected not only in butters of different origin [56] but also in yogurt or cheese [58]. The concentration of terpenes in SB was low compared to the other butters.

GB and CB samples presented the highest percentages of acids, with acetic and butanoic acids being the most representative. Short-chain fatty acids can originate not only from the hydrolysis of triglycerides but also from the degradation of lactose and amino acids [60], and they are commonly detected in butters and present low perception thresholds. In fact, butanoic acid is added to butter substitutes such as margarines to give them the aroma of butter [61].

Aliphatic hydrocarbons (alkanes) can be derived from both feed and lipid oxidation [62], but they contribute little to dairy product aroma because their odor thresholds are high [63]. However, alkanes serve as precursors for other aromatic compounds through various degradation pathways. Here, the extraction of alkanes showed a similar pattern to that previously reported in dairy products [64], and they were more abundant in low-fat butter made of cow’s milk (CLB) than in the other samples. Nonane was present in all the samples analyzed except for SB, and it showed the highest percentage of all the alkanes, being particularly high in the CLB samples.

Regarding alcohols, ethanol, which is the final product of glucose metabolism or amino acid degradation in milk, is believed to impact sweetness. However, its contribution to flavor is likely to be minimal due to its exceedingly high odor threshold [65]. Here, high percentages of this alcohol were detected in the GB and CSB samples, while 1-Nonanol and 2,3-butanediol were detected in all the samples analyzed, with an extremely high percentage of the latter in the CB and CSB samples.

The straight-chain aldehydes detected in dairy products are reported to be derived from the oxidation of PUFAs, although their origin can also derive from vegetable material transferred into milk [66]. In our study, the main aldehydes detected were hexanal (detected in all the samples except for CB) and tetradecanal. Hexanal is commonly found in the volatile profile of oxidized foods [67] and is produced through the autoxidation of linoleic acid. It has been selected as a marker for lipid oxidation in butter that is being stored [68]. Benzaldehyde, an aromatic aldehyde, was present in all the samples, but its percentage was particularly high in the CLB samples. This aldehyde can be generated through two different pathways: firstly, via the α-oxidation of phenylacetaldehyde, and secondly, through the ß-oxidation of cinnamic acid [69]. Its ability to give an aroma of bitter almonds to dairy products has also been reported [70].

Lactones are organic compounds produced when hydroxyacids undergo intramolecular esterification, resulting in the loss of water [65]. The lactones detected in this study were present in all the samples, and both caprolactone and octalactone have been previously detected in commercial butters [71]. The detection intensity of δ-octalactone seemed to be affected by the forage type [72] and also by the storage time [73].

Regarding aromatic hydrocarbons, toluene was only present in GB, CB, and CLB samples, while xylene was found only in SB and GB butters and in a very low concentration. Toluene was correlated with pasture-derived butter [74], while Xylene may be the result of carotenoid degradation, namely β-carotene degradation in the rumen, or possibly directly transferred from the feed [75]. Phenol was also detected in all the samples in this study, but the differences among the samples were not significant.

Esters are derived from the reaction between short- to medium-chain fatty acids and both primary and secondary alcohols that are obtained from lactose fermentation or from the catabolism of amino acids [60]. They usually provide sweet, fruity, and floral aromas to dairy products. The high percentage of ethyl butanoate in the CB and CSB samples was the most notable incidence regarding ester compounds.

While sulfur compounds can be generated through the breakdown of amino acids and vitamins, the primary source of these compounds is the degradation of methionine and cysteine [56]. The metabolic destination of compounds like dimethyl sulfide involves its oxidation into dimethyl sulfone, which can be passed into the milk used to produce butter. The percentage of dimethyl sulfone, the only sulfur compound detected in this study, was particularly high in the GB and CB samples. Dimethyl sulfone has been previously detected in milk [72] and butter produced using cow’s, sheep’s, or goat’s milk [56].

The multivariate data analysis of butter fatty acids and the percentages of volatile compounds showed clear differences between the butters by the species of the milk source and showed how different low-fat butter is to the rest. Based on this analysis, butters made from cow’s milk (excluding low-fat butter, which included more ingredients than milk fat) and goat’s milk showed a more aromatic profile associated with a higher level of unsaturated fats, compared to those made from sheep’s milk.

## 5. Conclusions

In this study, our approach involved analyzing the different versions of butter available on the market to highlight the different aspects that can affect their quality. Although significant differences were found in the samples analyzed for most of the parameters considered, it was their different fatty acid profiles (due to the different species of origin of the milk) that determined the main differences in terms of the product’s nutritional value, texture, volatile profile, and technological applications. The multivariate data analysis of butter fatty acids and volatile compound percentages showed a clear differentiation in the butters by the species of the milk source and emphasized how different low-fat butter is to the rest.

## Figures and Tables

**Figure 1 animals-13-03559-f001:**
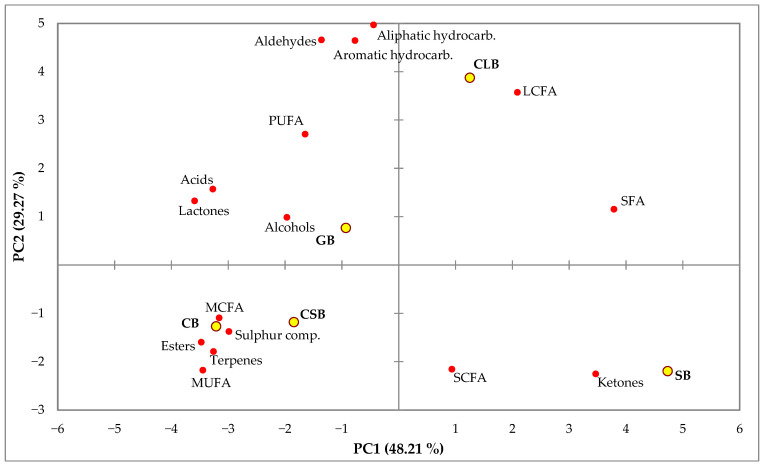
Principal component analysis plot representing the differentiation between samples of commercial butters based on grouped fatty acids and volatile compound chemical families.

**Table 1 animals-13-03559-t001:** Physicochemical parameters of commercial butters.

	SB	GB	CB	CSB	CLB	*p*-Value
Moisture (%)	15.89 ^b^ ± 0.73	13.55 ^c^ ± 0.10	15.63 ^bc^ ± 0.25	14.89 ^bc^ ± 0.60	58.89 ^a^ ± 0.29	***
SFC (%)	81.95 ^c^ ± 0.41	85.61 ^a^ ± 0.12	83.73 ^b^ ± 0.06	84.35 ^ab^ ± 0.38	37.21 ^d^ ± 0.63	***
Non-fat solids (%)	2.16 ^b^ ± 0.33	0.84 ^c^ ± 0.02	0.88 ^c^ ± 0.23	0.75 ^c^ ± 0.23	3.90 ^a^ ± 0.34	***
Chlorides (%)	1.03 ^b^ ± 0.05	1.17 ^ab^ ± 0.01	0.04 ^c^ ± 0.00	1.29 ^a^ ± 0.05	1.22 ^ab^ ± 0.11	***
Tritable acidity (%)	0.241 ^a^ ± 0.005	0.062 ^b^ ± 0.004	0.077 ^b^ ± 0.003	0.052 ^b^ ± 0.018	0.074 ^b^ ± 0.001	***
a_W_	0.915 ^b^ ± 0.001	0.909 ^b^ ± 0.007	0.928 ^b^ ± 0.010	0.902 ^b^ ± 0.012	0.986 ^a^ ± 0.002	***

^a–d^: Means with different superscript in the same row presented significant differences. SB = butter made from sheep’s milk; GB = butter made from goat’s milk; CB = unsalted butter made from cow’s milk; CSB = salted butter made from cow’s milk; CLB = low-fat butter made from cow’s milk. *p*-values: *** (*p* < 0.001). SFC = solid fat content; a_W_ = water activity.

**Table 2 animals-13-03559-t002:** Color parameters of commercial butters.

	SB	GB	CB	CSB	CLB	*p*-Value
*L**	89.27 ^ab^ ± 0.93	82.97 ^d^ ± 0.57	86.87 ^bc^ ± 0.94	85.70 ^cd^ ± 1.84	90.21 ^a^ ± 2.16	***
*a**	−3.17 ^c^ ± 0.19	−2.81 ^c^ ± 0.12	4.25 ^a^ ± 0.05	3.36 ^b^ ± 0.32	3.17 ^b^ ± 0.47	***
*b**	11.48 ^d^ ± 0.63	6.50 ^d^ ± 0.14	31.10 ^a^ ± 0.73	27.46 ^b^ ± 1.01	22.29 ^c^ ± 0.96	***
*C**	11.91 ^d^ ± 0.65	7.08 ^e^ ± 0.11	31.39 ^a^ ± 0.72	27.66 ^b^ ± 1.02	22.52 ^c^ ± 1.01	***
*h* ^0^	82.22 ^b^ ± 0.13	83.04 ^b^ ± 0.57	81.94 ^b^ ± 0.94	105.45 ^a^ ± 0.63	113.41 ^a^ ± 1.21	***
*WI*	65.96 ^e^ ± 0.30	68.81 ^d^ ± 0.42	75.36 ^c^ ± 0.39	83.94 ^a^ ± 0.37	81.56 ^b^ ± 0.54	***
*YI*	51.14 ^a^ ± 0.66	45.76 ^a^ ± 0.96	35.29 ^b^ ± 0.86	18.36 ^c^ ± 0.88	11.19 ^d^ ± 0.29	***
∆*E**				4.53 ± 1.79	9.80 ± 0.89	

^a–e^: Means with different superscript in the same row presented significant differences. SB = butter made from sheep’s milk; GB = butter made from goat’s milk; CB = unsalted butter made from cow’s milk; CSB = salted butter made from cow’s milk; CLB = low-fat butter made from cow’s milk. *p*-values: *** (*p* < 0.001). *h*^0^ = hue angle; *C** = chroma; *WI* = whiteness index; *YI* = yellowness index; Δ*E** color difference.

**Table 3 animals-13-03559-t003:** Textural properties of commercial butters.

	SB	GB	CB	CSB	CLB	Temp	BT	Temp × BT
Hardness (cutting force) (g)								
4 °C	233.7 ^a,yz^ ± 2.0	399.2 ^a,wx^ ± 29.6	335.8 ^a,xy^ ± 16.0	467.7 ^a,w^ ± 90.2	124.7 ^a,z^ ± 8.1	***	***	***
10 °C	102.6 ^b,y^ ± 9.9	255.6 ^b,v^ ± 12.8	189.4 ^b,x^ ± 6.0	231.2 ^b,w^ ± 2.1	58.2 ^b,z^ ± 1.0			
20 °C	9.9 ^c,z^ ± 0.2	20.7 ^c,y^ ± 2.0	43.4 ^c,w^ ± 4.6	36.3 ^c,x^ ± 0.6	9.9 ^c,z^ ± 0.7			
Hardness (penetration force) (g)								
4 °C	300.1 ^a,y^ ± 20.9	517.1 ^a,x^ ± 43.8	569.7 ^a,wx^ ± 36.3	655.7 ^a,w^ ± 60.5	136.5 ^a,z^ ± 29.7	***	***	***
10 °C	155.3 ^b,y^ ± 8.4	295.6 ^b,x^ ± 42.2	308.8 ^b,x^ ± 9.2	333.4 ^b,x^ ± 6.9	80.2 ^b,z^ ± 3.9			
20 °C	10.2 ^c,z^ ± 0.04	38.4 ^c,y^ ± 2.7	86.1 ^c,x^ ± 3.3	87.7 ^c,x^ ± 5.9	18.3 ^c,z^ ± 0.6			
Adhesiveness (g)								
4 °C	48.9 ± 23.5	14.3 ^b^ ± 11.2	39.8 ^b^ ± 15.3	32.6 ^b^ ± 21.4	56.1 ^a^ ± 22.4	***	**	***
10 °C	44.9 ^z^ ± 28.6	22.4 ^b,z^ ± 1.0	252.9 ^a,y^ ± 73.4	168.3 ^a,yz^ ± 140.7	55.1 ^a,z^ ± 0.11			
20 °C	14.3 ^z^ ± 3.1	60.2 ^a,z^ ± 7.2	226.4 ^a,y^ ± 48.9	228.4 ^a,y^ ± 11.2	14.3 ^b,z^ ± 0.04			

^a–c^: Means with different superscript in the same row presented significant differences. ^v–z^: Means with different superscript in the same column presented significant differences. SB = butter made from sheep’s milk; GB = butter made from goat’s milk; CB = unsalted butter made from cow’s milk; CSB = salted butter made from cow’s milk; CLB = low-fat butter made from cow’s milk. Temp = temperature; BT = butter type. *p*-values: ** (*p* < 0.01), *** (*p* < 0.001).

**Table 4 animals-13-03559-t004:** Fatty acids identified (expressed as % of the total fatty acid detected) in commercial butters.

Fatty Acids and Indexes	SB	GB	CB	CSB	CLB	*p*-Value
C4:0	4.18 ^a^ ± 0.04	4.11 ^a^ ± 0.08	2.77 ^b^ ± 0.05	2.39 ^c^ ± 0.09	1.72 ^d^ ± 0.00	***
C6:0	3.84 ^a^ ± 0.05	3.77 ^a^ ± 0.10	2.53 ^b^ ± 0.13	2.18 ^c^ ± 0.04	1.45 ^d^ ± 0.02	***
C8:0	3.43 ^a^ ± 0.07	3.46 ^a^ ± 0.02	2.31 ^b^ ± 0.02	2.07 ^c^ ± 0.00	1.52 ^d^ ± 0.02	***
C10:0	6.53 ^a^ ± 0.13	5.16 ^b^ ± 0.13	3.06 ^c^ ± 0.04	2.51 ^d^ ± 0.02	2.34 ^d^ ± 0.04	***
C10:1	0.23 ^a^ ± 0.02	0.13 ^b^ ± 0.00	0.30 ^a^ ± 0.02	0.23 ^a^ ± 0.01	0.25 ^a^ ± 0.03	**
C12:0	4.75 ^a^ ± 0.04	4.16 ^bc^ ± 0.12	3.81 ^c^ ± 0.14	3.51 ^cd^ ± 0.12	2.81 ^d^ ± 0.07	***
C12:1	0.13 ^cd^ ± 0.01	0.19 ^b^ ± 0.00	0.11 ^d^ ± 0.01	0.18 ^bc^ ± 0.01	0.32 ^a^ ± 0.03	***
C14:0	10.32 ^c^ ± 0.04	9.20 ^d^ ± 0.05	12.29 ^a^ ± 0.10	11.77 ^b^ ± 0.13	7.02 ^e^ ± 0.03	***
iC15:0	0.17 ^b^ ± 0.00	0.18 ^b^ ± 0.01	0.25 ^a^ ± 0.03	0.18 ^b^ ± 0.01	0.24 ^a^ ± 0.00	**
C14:1	0.45 ^d^ ± 0.00	0.27 ^e^ ± 0.03	1.40 ^a^ ± 0.06	1.22 ^b^ ± 0.04	1.03 ^c^ ± 0.05	***
aC15:0	0.11 ^c^ ± 0.01	0.23 ^b^ ± 0.01	0.00 ^d^ ± 0.00	0.08 ^c^ ± 0.01	0.32 ^a^ ± 0.01	***
C15:0	0.78 ^c^ ± 0.02	0.55 ^d^ ± 0.04	1.19 ^a^ ± 0.01	1.10 ^b^ ± 0.00	0.84 ^c^ ± 0.01	***
iC16:0	0.11 ^ab^ ± 0.01	0.13 ^a^ ± 0.01	0.00 ^c^ ± 0.00	0.07 ^b^ ± 0.02	0.11 ^ab^ ± 0.00	***
C15:1	0.13 ^d^ ± 0.01	0.14 ^cd^ ± 0.02	0.25 ^ab^ ± 0.01	0.19 ^bc^ ± 0.02	0.29 ^a^ ± 0.02	***
C16:0	27.97 ^d^ ± 0.19	31.05 ^c^ ± 0.05	32.83 ^b^ ± 0.40	33.64 ^b^ ± 0.66	36.48 ^a^ ± 0.08	***
C16:1	1.02 ^b^ ± 0.08	0.77 ^c^ ± 0.01	1.66 ^a^ ± 0.09	1.52 ^a^ ± 0.03	0.95 ^bc^ ± 0.02	***
iC17:0	0.23 ^b^ ± 0.04	0.19 ^b^ ± 0.00	0.28 ^ab^ ± 0.00	0.33 ^a^ ± 0.04	0.27 ^ab^ ± 0.01	*
C17:0	0.43 ^a^ ± 0.04	0.31 ^b^ ± 0.01	0.45 ^a^ ± 0.00	0.43 ^a^ ± 0.03	0.26 ^b^ ± 0.00	**
C17:1	0.20 ± 0.02	0.22 ± 0.01	0.14 ± 0.01	0.22 ± 0.01	0.11 ± 0.15	ns
C18:0	11.52 ^d^ ± 0.34	15.86 ^b^ ± 0.27	10.93 ^d^ ± 0.13	12.90 ^c^ ± 0.03	21.81 ^a^ ± 0.17	***
C18:1n9t	1.45 ^a^ ± 0.07	1.17 ^b^ ± 0.04	0.82 ^c^ ± 0.04	0.32 ^d^ ± 0.06	0.83 ^c^ ± 0.04	***
C18:1n9c	16.62 ^b^ ± 0.28	15.37 ^c^ ± 0.29	19.09 ^a^ ± 0.02	18.85 ^a^ ± 0.31	14.95 ^c^ ± 0.07	***
C18:2n6t	0.17 ^b^ ± 0.00	0.21 ^ab^ ± 0.02	0.14 ^b^ ± 0.01	0.27 ^a^ ± 0.03	0.15 ^b^ ± 0.03	*
C18:2n6c	2.93 ^b^ ± 0.01	2.01 ^d^ ± 0.05	2.49 ^c^ ± 0.10	2.80 ^b^ ± 0.02	3.17 ^a^ ± 0.01	***
C18:3n6g	0.08 ± 0.01	0.07 ± 0.00	0.05 ± 0.01	0.06 ± 0.01	0.07 ± 0.03	ns
C20:0	0.17 ^a^ ± 0.01	0.16 ^a^ ± 0.01	0.09 ^b^ ± 0.01	0.14 ^ab^ ± 0.03	0.11 ^ab^ ± 0.01	*
C18:3n3a	1.52 ^a^ ± 0.09	0.21 ^b^ ± 0.02	0.29 ^b^ ± 0.02	0.33 ^b^ ± 0.04	0.16 ^b^ ± 0.01	***
C20:1n9	0.47 ^b^ ± 0.00	0.69 ^a^ ± 0.00	0.35 ^bc^ ± 0.05	0.37 ^bc^ ± 0.09	0.26 ^c^ ± 0.01	**
C20:3n6	0.05 ^b^ ± 0.00	0.06 ^b^ ± 0.00	0.12 ^a^ ± 0.01	0.15 ^a^ ± 0.02	0.17 ^a^ ± 0.02	**
SFA	74.55 ^c^ ± 0.16	78.51 ^a^ ± 0.21	72.79 ^d^ ± 0.18	73.31 ^d^ ± 0.35	77.28 ^b^ ± 0.15	***
MUFA	20.70 ^c^ ± 0.22	18.95 ^d^ ± 0.25	24.11 ^a^ ± 0.07	23.09 ^b^ ± 0.38	19.00 ^d^ ± 0.11	***
PUFA	4.75 ^a^ ± 0.07	2.54 ^d^ ± 0.04	3.09 ^c^ ± 0.11	3.60 ^b^ ± 0.02	3.72 ^b^ ± 0.04	***
SCFA	17.97 ^a^ ± 0.19	16.49 ^b^ ± 0.08	10.67 ^c^ ± 0.23	9.15 ^d^ ± 0.15	7.02 ^e^ ± 0.01	***
MCFA	44.25 ^d^ ± 0.27	45.39 ^c^ ± 0.13	50.48 ^a^ ± 0.02	50.41 ^a^ ± 0.36	47.61 ^b^ ± 0.32	***
LCFA	11.90 ^d^ ± 0.62	16.17 ^b^ ± 0.05	11.24 ^d^ ± 0.19	13.26 ^c^ ± 0.33	22.05 ^a^ ± 0.00	***
n-3	3.23 ^b^ ± 0.02	2.34 ^d^ ± 0.06	2.80 ^c^ ± 0.09	3.28 ^b^ ± 0.02	3.56 ^a^ ± 0.05	***
n-6	1.52 ^a^ ± 0.09	0.21 ^b^ ± 0.02	0.29 ^b^ ± 0.02	0.33 ^b^ ± 0.04	0.16 ^b^ ± 0.01	***
n-9	16.62 ^b^ ± 0.28	15.37 ^c^ ± 0.29	19.09 ^a^ ± 0.02	18.85 ^a^ ± 0.31	14.95 ^c^ ± 0.07	***
Atherogenic index	3.58 ^b^ ± 0.04	3.80 ^a^ ± 0.06	3.56 ^b^ ± 0.02	3.65 ^ab^ ± 0.06	3.54 ^b^ ± 0.01	**
Thrombogenic index	4.01 ^d^ ± 0.01	5.23 ^b^ ± 0.09	4.13 ^cd^ ± 0.01	4.39 ^c^ ± 0.10	5.77 ^a^ ± 0.06	***
Spreadability index	1.68 ^c^ ± 0.04	2.02 ^b^ ± 0.04	1.72 ^c^ ± 0.06	1.79 ^c^ ± 0.06	2.44 ^a^ ± 0.02	***

^a–e^: Means with different superscript in the same row presented significant differences. SB = butter made from sheep’s milk; GB = butter made from goat’s milk; CB = unsalted butter made from cow’s milk; CSB = salted butter made from cow’s milk; CLB = low-fat butter made from cow’s milk. SFA: Σ saturated fatty acids; MUFA: Σ monounsaturated fatty acid; PUFA: Σ polyunsaturated fatty acid; SCFA (short-chain fatty acids): C4:0 to C10:0; MCFA (medium chain fatty acids): C11:0 to C17:0; LCFA (long-chain fatty acids): >C18:0; n-3: Σ omega-3 fatty acids; n-6: Σ omega-6 fatty acids; n-9: Σ omega-9 fatty acids. *p*-values: ns: not significant; * (*p* < 0.05), ** (*p* < 0.01), *** (*p* < 0.001).

**Table 5 animals-13-03559-t005:** Volatile compound species identified (expressed as % of the total volatile organic compounds detected) in commercial butters.

Volatile Compounds	LRI	SB	GB	CB	CSB	CLB	*p*-Value
**Ketones**							
Acetone	859	1.32 ^c^ ± 1.32	4.72 ^bc^ ± 1.15	18.61 ^ab^ ± 4.58	5.32 ^bc^ ± 1.74	20.06 ^a^ ± 10.93	**
2-Pentanone	990	5.23 ± 5.00	0.96 ± 0.50	0.00 ± 0.00	0.00 ± 0.00	0.00 ± 0.00	ns
Heptan-2-one	1186	30.97 ^a^ ± 0.15	2.75 ^c^ ± 0.51	2.75 ^c^ ± 0.39	2.00 ^c^ ± 0.55	12.55 ^b^ ± 0.14	***
3-Hydroxy 2-butanone (acetoin)	1299	0.65 ^c^ ± 0.25	25.4 ^a^ ± 2.36	15.69 ^b^ ± 2.96	31.94 ^a^ ± 3.77	6.28 ^c^ ± 1.37	***
2-Nonanone	1396	47.57 ^a^ ± 4.08	8.48 ^b^ ± 0.47	2.04 ^b^ ± 0.02	2.01 ^b^ ± 0.05	5.14 ^b^ ± 4.32	***
8-Nonen-2-one	1453	2.92 ^a^ ± 0.10	0.70 ^b^ ± 0.01	0.98 ^b^ ± 0.21	0.72 ^b^ ± 0.02	0.00 ^c^ ± 0.00	***
2-Undecanone	1613	2.70 ^a^ ± 0.24	1.67 ^b^ ± 0.10	0.26 ^c^ ± 0.03	0.11 ^c^ ± 0.02	0.00 ^c^ ± 0.00	***
Acetophenone	1672	0.27 ± 0.08	0.39 ± 0.17	0.48 ± 0.14	0.30 ± 0.17	0.43 ± 0.42	ns
Total ketones (8)		91.63	45.07	40.81	42.40	44.46	
**Terpenes**							
α-Pinene	1024	0.00 ^b^ ± 0.00	7.19 ^a^ ± 2.12	0.00 ^b^ ± 0.00	0.00 ^b^ ± 0.00	0.00 ^b^ ± 0.00	***
p-Menth-3-ene	1129	0.00 ^b^ ± 0.00	0.00 ^b^ ± 0.00	1.40 ^a^ ± 0.16	0.00 ^b^ ± 0.00	0.00 ^b^ ± 0.00	***
Limonene	1202	0.00 ^b^ ± 0.00	0.66 ^b^ ± 0.08	7.16 ^a^ ± 1.89	9.95 ^a^ ± 1.95	3.25 ^b^ ± 0.67	***
p-Cymene	1277	0.00 ^d^ ± 0.00	0.00 ^d^ ± 0.00	11.96 ^b^ ± 0.64	15.11 ^a^ ± 0.64	1.26 ^c^ ± 0.42	***
4-Carene	1289	1.13 ^a^ ± 0.05	0.00 ^b^ ± 0.00	1.42 ^a^ ± 0.22	1.19 ^a^ ± 0.34	0.00 ^b^ ± 0.00	***
Total terpenes (5)		1.13	7.85	21.94	26.25	4.51	
**Acids**							
Acetic acid	1477	0.13 ^c^ ± 0.01	7.09 ^ab^ ± 0.11	10.16 ^a^ ± 1.72	5.92 ^ab^ ± 1.40	4.09 ^bc^ ± 3.93	***
Butanoic acid	1654	2.85 ± 0.17	4.39 ± 1.50	2.74 ± 0.13	1.60 ± 0.47	2.28 ± 1.78	ns
2-Methylbutanoic acid	1695	0.24 ^b^ ± 0.03	0.27 ^ab^ ± 0.09	0.38 ^a^ ± 0.05	0.10 ^c^ ± 0.01	0.00 ^c^ ± 0.00	***
Hexanoic acid	1873	0.30 ^b^ ± 0.08	3.00 ^a^ ± 1.19	1.22 ^b^ ± 0.38	1.04 ^b^ ± 0.22	3.26 ^a^ ± 0.05	***
Octanoic acid	2088	0.11 ^c^ ± 0.01	0.00 ^c^ ± 0.00	0.80 ^ab^ ± 0.42	0.50 ^bc^ ± 0.06	1.07 ^a^ ± 0.03	***
Total acids (5)		3.63	14.75	15.30	9.16	10.70	
**Aliphatic hydrocarbons**							
Nonane	1001	0.00 ^c^ ± 0.00	8.70 ^b^ ± 1.81	3.25 ^bc^ ± 2.89	3.81 ^bc^ ± 1.77	17.59 ^a^ ± 5.00	***
Undecane	1183	0.00 ± 0.00	1.55 ± 1.50	0.00 ± 0.00	0.00 ± 0.00	0.88 ± 0.30	ns
Tetradecane	1504	0.58 ^bc^ ± 0.20	0.34 ^c^ ± 0.19	1.34 ^a^ ± 0.08	0.35 ^c^ ± 0.31	1.19 ^ab^ ± 0.33	***
Hexadecane	1608	0.04 ^c^ ± 0.01	0.51 ^bc^ ± 0.12	1.23 ^ab^ ± 0.58	0.50 ^bc^ ± 0.21	1.82 ^a^ ± 0.36	***
Heptadecane	1709	0.06 ^c^ ± 0.01	0.55 ^bc^ ± 0.07	0.88 ^ab^ ± 0.19	0.49 ^bc^ ± 0.10	1.14 ^a^ ± 0.35	***
Nonadecane	1912	0.04 ^b^ ± 0.01	0.23 ^b^ ± 0.08	0.26 ^b^ ± 0.24	0.20 ^b^ ± 0.04	0.69 ^a^ ± 0.03	***
Total alip. hydrocarb. (6)		0.72	11.88	6.96	5.35	23.31	
**Alcohols**							
Ethanol	941	0.30 ^b^ ± 0.14	7.77 ^a^ ± 4.90	0.00 ^b^ ± 0.00	4.73 ^ab^ ± 1.25	1.94 ^ab^ ± 0.91	**
1-Pentanol	1263	0.14 ^c^ ± 0.10	0.53 ^b^ ± 0.12	0.00 ^c^ ± 0.00	1.23 ^a^ ± 0.13	0.70 ^b^ ± 0.14	***
1-Nonanol	1497	0.06 ^b^ ± 0.02	0.59 ^ab^ ± 0.30	0.66 ^ab^ ± 0.42	0.56 ^ab^ ± 0.21	0.88 ^a^ ± 0.06	*
Octan1-ol	1569	0.34 ^a^ ± 0.14	0.25 ^ab^ ± 0.16	0.00 ^b^ ± 0.00	0.00 ^b^ ± 0.00	0.22 ^ab^ ± 0.08	**
2,3- Butanediol	1593	0.46 ± 0.09	0.69 ± 0.02	2.48 ± 2.29	1.99 ± 0.40	0.49 ± 0.12	ns
Total alcohols (5)		1.30	9.83	3.14	8.51	4.23	
**Aldehydes**							
Hexanal	1086	0.50 ^c^ ± 0.04	1.84 ^b^ ± 0.04	0.00 ^d^ ± 0.00	1.49 ^b^ ± 0.01	2.53 ^a^ ± 0.39	***
Benzaldehyde	1543	0.28 ^c^ ± 0.21	0.92 ^bc^ ± 0.23	0.98 ^b^ ± 0.35	0.72 ^bc^ ± 0.17	1.67 ^a^ ± 0.21	***
Tetradecanal	1727	0.14 ^b^ ± 0.00	1.42 ^a^ ± 0.38	2.02 ^a^ ± 0.33	1.28 ^a^ ± 0.20	2.06 ^a^ ± 0.56	***
Octadecanal	1942	0.10 ± 0.03	0.37 ± 0.09	0.47 ± 0.16	0.37 ± 0.14	0.62 ± 0.48	ns
Total aldehydes (4)		1.02	4.55	3.47	3.86	6.88	
**Lactones**							
δ-Caprolactone	1820	0.04 ^c^ ± 0.01	1.27 ^b^ ± 0.15	1.99 ^a^ ± 0.38	1.28 ^b^ ± 0.28	1.00 ^b^ ± 0.05	***
δ-Octalactone	1996	0.05 ^b^ ± 0.01	0.53 ^ab^ ± 0.21	0.97 ^a^ ± 0.05	0.52 ^ab^ ± 0.13	0.90 ^a^ ± 0.41	**
Total lactones (2)		0.09	1.80	2.96	1.80	1.90	
**Aromatic hydrocarbons**							
Toluene	1047	0.00 ^b^ ± 0.00	1.84 ^ab^ ± 1.25	1.05 ^ab^ ± 0.01	0.00 ^b^ ± 0.00	2.78 ^a^ ± 1.14	**
p-Xylene	1138	0.23 ^a^ ± 0.01	0.24 ^a^ ± 0.14	0.00 ^b^ ± 0.00	0.00 ^b^ ± 0.00	0.00 ^b^ ± 0.00	***
Phenol	2039	0.04 ± 0.01	0.32 ± 0.11	0.71 ± 0.35	0.46 ± 0.19	0.79 ± 0.46	ns
Total arom. hydrocarb. (3)		0.24	2.40	1.76	0.46	3.57	
**Esters**							
Ethyl butanoate	1037	0.00 ^c^ ± 0.00	0.00 ^c^ ± 0.00	1.89 ^a^ ± 0.33	1.33 ^b^ ± 0.29	0.00 ^c^ ± 0.00	***
Butyl hexanoate	1412	0.06 ^b^ ± 0.01	0.30 ^a^ ± 0.10	0.00 ^b^ ± 0.00	0.00 ^b^ ± 0.00	0.00 ^b^ ± 0.00	***
2-Phenylethyl acetate	1862	0.00 ± 0.00	0.30 ± 0.20	0.32 ± 0.25	0.14 ± 0.03	0.00 ± 0.00	ns
Methyl pentadecanoate	2028	0.02 ^b^ ± 0.01	0.17 ^ab^ ± 0.10	0.33 ^ab^ ± 0.20	0.21 ^ab^ ± 0.04	0.43 ^a^ ± 0.16	*
Total esters (4)		0.08	0.77	2.54	1.68	0.43	
**Suphur compounds**							
Dimethylsulfone	1934	0.11 ^b^ ± 0.02	1.12 ^a^ ± 0.43	1.12 ^a^ ± 0.13	0.54 ^b^ ± 0.04	0.00 ^b^ ± 0.00	***
Total sulphur comp. (1)		0.11	1.12	1.12	0.54	0.00	

LRI: linear retention index. ^a–d^: Means with different superscript in the same row presented significant differences. SB = butter made from sheep’s milk; GB = butter made from goat’s milk; CB = unsalted butter made from cow’s milk; CSB = salted butter made from cow’s milk; CLB = low-fat butter made from cow’s milk. *p*-values: ns: not significant; * (*p* < 0.05), ** (*p* < 0.01), *** (*p* < 0.001).

## Data Availability

Data are contained within the article.

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
