# Peer review of "Butter from Different Species: Composition and Quality Parameters of Products Commercialized in the South of Spain"

_animals, 2023, doi:10.3390/ani13223559_

Round 1

Reviewer 1 Report

Comments and Suggestions for Authors

The objective of the present study was to investigate the Composition and quality parameters of Butter from different species commercialized in the south of Spain.
The research is interesting and the authors made a number of analytical determinations to better highlight the differences between the parameters of Butter from different species.
However I therefore have to point out some comments:
Regarding the introduction, I think it is necessary the authors  to add information that refers to the production and consumption of butter for the three types of milk, both in the country and in Europe, as well as the current legislation on butter products.
In the materials and methods section, I consider in principle the number of samples to be limited. A more detailed description of the methods would also be useful regardless of whether they are ISO  methods
Finally in the results, It would be even better to compare the results of chemical analysis with the composition indicated on the label of the samples (butter) examined and to state in the results or conclusions whether the values of the chemical composition (moisture, fat, salt, etc.) are  similar.

Author Response

Dear reviewer,

We have found all your comments and suggestions very helpful and we think that they greatly contribute to improving the quality of our manuscript. Please find attached a file with the responses to your recommendations.

Regards,

The authors

Reviewer 2 Report

Comments and Suggestions for Authors

Comments and suggestions to the authors of the paper.

I have certain comments about the number of analyses performed (only one sample of each type of butter in three replicates). The number of records allows statistical analysis, while due to the large influence of environmental and genetic factors on the composition of milk, a much larger number of samples from different production batches should have been analyzed. The results obtained for several different samples would have greater scientific value.

Ruminant milk is the main source of CLA isomers in the human diet. Chromatographic analyses determined the content of two conjugated dienes of linoleic acid, while their content was not reported in butter.

The results should be analyzed and confronted independently in the context of the species of milk from which the butter was produced (sheep, goat, cow) and of the technological processes used (normal, salted, low-fat). Therefore some information of the text in the section "Result and disscusion" need to be rewritten and supplemented in this regard.

In several places, the description of results in the text is incompatible with the data in the table.

In the Table 3 incorrectly marks statistical differences between groups. Please check Table 5 for statistical differences. Most of the tables or legends to the tables need minor editorial corrections.

More references should be corrected as per the journal requirements (for example abbreviated journal name).

Some detailed comments:

Ln 15: A range of butter? I would suggest rather, several types of butter.

Ln 22-24: The composition of the original milk (…).In the study did not examine milk only butter obtained from the milk of different animal species. Please reedit this sentence.

Ln 100: (Texture Technologies Corp., Scarsdale, NY, USA)

Ln 113-114: Agilent 6890 Network GS System (Agilent, Santa Clara, CA, USA)

Ln 198-199: SB contains more total protein than GB and CB and, consequently, higher acidity values. The study did not examine total protein content. In addition to protein, the non-fat solid includes lactose and ash. No differences in water activity were found between the SB group and the GB and CB groups. It is important to be precise.

Ln 208:  CLB presented significantly higher aW values (p > 0.05) – please see table.

Ln 221: I would suggest either using the full names for SFC and aw in the table, or putting an explanation of the abbreviations in the legend. I would suggest leaving only P-values in the legend: *** (p < 0.001), since no differences were found at other levels of significance.

Ln 262: I would suggest including an explanation of the abbreviations in the legend to Table 2:  hue angle (h0), color intensity or chroma (C*), whiteness index (WI), yellowness index (YI) and color difference (ΔE*). The description of the table should be precise enough to allow understand its contents without knowing the text of the work. I would suggest in the legend leaving only P-values: *** (p < 0.001).

Ln 306: Incorrectly in relation to the information of the posted legend, the differences between the groups were presented. I would suggest leaving only P-values: ** (p < 0.01), *** (p < 0.001) in the legend.

Ln 315-316: „Overall, 25 of these fatty acids varied significantly (p < 0.05) among butters”. The table shows that 27 FAs.

Ln 318: (SFA, MUFA and PUFA)

Ln 326: „to their high lauric (C12:0)” – this is not apparent from the data in the table.

Ln 338-339: This is not apparent from the data in the table

Ln 361: lauric acid (C12:0, 2.81–4.75%).

Ln 378, 386: Atherogenic index

Ln 379: Missing or unnecessary spaces. I would suggest leaving only: ** (p < 0.01), *** (p < 0.001).

Ln 394: Trombogenic index

Ln 375: PUFA instead of PFA and MUFA instead of MFA

Ln 505: MUFA and PUFA

Author Response

(The authors gave the same response as above.)
